# Minimally Invasive Surgery and Surgical Volume-Specific Survival and Perioperative Outcome: Unmet Need for Evidence in Gynecologic Malignancy

**DOI:** 10.3390/jcm10204787

**Published:** 2021-10-19

**Authors:** Shinya Matsuzaki, Maximilian Klar, Erica J. Chang, Satoko Matsuzaki, Michihide Maeda, Renee H. Zhang, Lynda D. Roman, Koji Matsuo

**Affiliations:** 1Department of Gynecology, Osaka International Cancer Institute, Osaka 541-8567, Japan; maeda.rf@gmail.com; 2Division of Gynecologic Oncology, Department of Obstetrics and Gynecology, University of Southern California, Los Angeles, CA 90033, USA; erica.chang@med.usc.edu (E.J.C.); lroman@med.usc.edu (L.D.R.); koji.matsuo@gmail.com (K.M.); 3Department of Obstetrics and Gynecology, Osaka University Graduate School of Medicine, Osaka 565-0871, Japan; 4Department of Obstetrics and Gynecology, University of Freiburg, 79085 Freiburg, Germany; maximilian.klar@uniklinik-freiburg.de; 5Department of Obstetrics and Gynecology, Osaka General Medical Center, Osaka 558-8558, Japan; satoko_tsuru@yahoo.co.jp; 6Keck School of Medicine, University of Southern California, Los Angeles, CA 90033, USA; reneezha@usc.edu; 7Norris Comprehensive Cancer Center, University of Southern California, Los Angeles, CA 90033, USA

**Keywords:** minimally invasive surgery, surgical volume, volume–outcome relationship, survival, gynecologic malignancy, systematic review

## Abstract

This study examined the effect of hospital surgical volume on oncologic outcomes in minimally invasive surgery (MIS) for gynecologic malignancies. The objectives were to assess survival outcomes related to hospital surgical volume and to evaluate perioperative outcomes and examine non-gynecologic malignancies. Literature available from the PubMed, Scopus, and the Cochrane Library databases were systematically reviewed. All surgical procedures including gynecologic surgery with hospital surgical volume information were eligible for analysis. Twenty-three studies met the inclusion criteria, and nine gastro-intestinal studies, seven genitourinary studies, four gynecological studies, two hepatobiliary studies, and one thoracic study were reviewed. Of those, 11 showed a positive volume–outcome association for perioperative outcomes. A study on MIS for ovarian cancer reported lower surgical morbidity in high-volume centers. Two studies were on endometrial cancer, of which one showed lower treatment costs in high-volume centers and the other showed no association with perioperative morbidity. Another study examined robotic-assisted radical hysterectomy for cervical cancer and found no volume–outcome association for surgical morbidity. There were no gynecologic studies examining the association between hospital surgical volume and oncologic outcomes in MIS. The volume–outcome association for oncologic outcome in gynecologic MIS is understudied. This lack of evidence calls for further studies to address this knowledge gap.

## 1. Introduction

Recently, minimally invasive surgery (MIS) has become a common procedure in benign and malignant diseases [1]. Numerous observational studies have documented the feasibility of MIS in various malignant diseases [2]. Compared to open surgery, MIS reduces the perioperative morbidity and duration of hospital admission [3].

The perioperative and oncologic outcomes of any surgery depend on a multitude of factors, the most influential being the tumor and patient characteristics [4]. There is growing recognition that factors related to the hospital system, such as the hospital surgical volume and surgeon volume, may also affect treatment [4]. While the association among hospital surgical volume, surgeon volume, and perioperative outcomes is well-established for open surgeries in malignant diseases, the association with MIS is less robust [5,6].

In MIS, hospital surgical volume may positively correlate to improved perioperative and oncologic outcomes because (i) MIS is more technically demanding than open surgery, (ii) robotic-assisted MIS for malignant disease has been recently introduced [7], and (iii) MIS is less frequently performed in complex surgeries for malignant diseases [8]. However, the volume–outcome association for perioperative and oncologic outcomes in MIS for malignancies is understudied; thus, little is known regarding the impact of hospital surgical volume on these cases. This study aimed to examine the survival effect of hospital surgical volume on MIS for gynecologic malignancies.

## 2. Materials and Methods

### 2.1. Approach for Systematic Literature Review

A systematic review was performed to determine the effect of hospital surgical volume on perioperative and oncologic outcomes in cases of malignant diseases, including gynecologic cancer treated with MIS. To review the current status of the case volume–outcome association for perioperative and survival outcomes in MIS, our analysis was not restricted to gynecologic surgery.

### 2.2. Article Retrieval

We conducted a systematic search of articles published through 30 June 2020 using the PubMed, Scopus, and Cochrane Central Register of Controlled Trials databases, as performed in our previous study [9,10,11,12]. We reviewed articles according to the Preferred Reporting Items for Systematic Reviews and Meta-Analyses guidelines [13,14]. Studies were identified by screening the titles, abstracts, and full texts of relevant articles, as previously described. All abstracts were screened by Sh.M.

The following terms were applied in the PubMed, Scopus, and the Cochrane databases to identify studies on MIS (Medical Subject Headings (MeSH) terms were used in the PubMed and Cochrane database search): minimally invasive surgery (MeSH) OR “minimally invasive” OR “endoscopic” OR “endoscope” OR “laparoscopic” OR “laparoscope” OR “robot*” OR “robotic.” Studies investigating the effect of annual surgical volume on MIS were then identified from this list using the following keywords: hospitals, high-Volume (MeSH) OR hospitals, low-volume (MeSH) OR “high volume center” OR “high volume institution” OR “high volume hospital” OR “high volume facility” OR “low volume center” OR “low volume institution” OR “low volume hospital” OR “low volume facility” OR “surgical volume” OR “treatment volume” OR “procedure volume” OR “care volume” OR “hospital volume” OR “case volume” OR “center volume” OR “facility volume” OR “annual case” OR “annual volume” OR “annual number” OR “health facility size” OR “hospital size” OR “clinic size” OR “provider volume.” To determine additional related studies, the references of related articles were also reviewed.

### 2.3. Study Selection

The inclusion criteria were set according to the Population/Intervention/Comparison/Outcomes (PICO) framework, as shown in Appendix A. Studies were included if they met the following criteria: (1) volume–outcome was examined according to the hospital’s annual surgical volume (not surgeon volume and surgeon experience); (2) patients were treated with minimally invasive oncologic surgery; and (3) surgical complications, cost, hospital mortality, the rate of laparotomy conversion, overall survival, and progression-free survival were assessed.

The exclusion criteria were as follows: (1) insufficient information about the hospital’s annual volume; (2) inadequate information on surgical complications, survival, or recurrence; (3) hospital surgical volume was assessed by summing the MIS and open surgery cases; (4) not written in English; and (5) conference abstracts, case reports, case series, and reviews.

### 2.4. Data Extraction

Data were extracted by Sh.M., and the following variables were recorded: surgery type, cancer type, the type of MIS, year of study, first author’s name, study location, number of included cases, the definition of high-volume center, and outcomes of interest (surgical and oncologic outcomes). The entered data were double-checked by the review author (Sa.M.).

### 2.5. Outcome Measures Analysis

The primary objective of the study was to assess the association between hospital surgical volume and survival outcome in MIS for gynecologic malignancies. Two secondary objectives were also examined. First, perioperative outcome, which included surgical morbidity, mortality, cost, length of hospital stay, rate of laparotomy conversion, and rate of positive surgical margins, was examined. Second, the study population of non-gynecologic malignancies was examined.

A risk of bias assessment was performed using the Risk Of Bias In Non-randomized Studies-of Interventions (ROBINS-I) tool, as previously performed [15,16,17,18,19].

In the sensitivity analysis, the effect of hospital surgical volume on the rates of infection, re-operation, the length of hospital stay, and the cost of the hospital stay was determined. We also determined whether the hospital surgical volume was analyzed according to patient age.

## 3. Results

### 3.1. Study Selection

A total of 3012 studies were examined, and 23 comprising 293,159 minimally invasive oncologic surgeries met the inclusion criteria and were used for the descriptive analysis [4,5,6,7,8,20,21,22,23,24,25,26,27,28,29,30,31,32,33,34,35,36,37]. The study selection schema is shown in Figure 1.

### 3.2. Study Characteristics

The metadata of the evaluated studies are shown in Table 1 and Appendix A.

### 3.3. Risk of Bias of Included Studies

The risk of bias assessment for the comparative studies demonstrated a possible low publication bias in 1 study, moderate publication bias in 19 studies, and severe publication bias in the other 3 studies (Appendix A).

Among the 23 studies, gastrointestinal surgery was the most common surgery type (nine studies) [20,21,22,23,24,25,26,27,28], followed by genitourinary (seven studies) [30,31,32,33,34,35,36], gynecologic (four studies) [4,5,6,7], hepatobiliary (two studies) [8,29], and thoracic (one study) surgeries [37]. In the gynecologic surgery group, two studies reported the outcomes of minimally invasive hysterectomy in endometrial cancer [4,6]. One study examined minimally invasive radical hysterectomy in cervical cancer [7], and one study examined minimally invasive oophorectomy in ovarian cancer [5].

Studies included in this review were published from 2009 to 2020. The study duration corresponding to these studies ranged from 1997 to 2015, but 22 of the 23 studies used a starting point of the 2000s [4,5,6,7,8,20,21,22,23,24,25,26,27,29,30,31,32,33,34,35,36,37]. The majority of studies were from the United States (69.6%) [4,5,6,7,8,20,21,24,25,29,30,32,33,34,35,37], followed by Europe (17.4%) [22,28,31,36] and Japan (13.0%) [23,26,27].

### 3.4. Definition of High-Volume Center

Among the 23 studies, the median hospital surgical volume for high-volume centers was 37.5 cases per year (Figure 2).

The definition of high hospital surgical volume varied across the studies (Table 1). The cutoff designating a high-volume center ranged from 2 to 219 surgeries a year. Three (13.0%) studies used the top decile cutoff [5,7,20], six (26.1%) used the top quartile cutoff [21,24,31,32,35,37], and two (8.7%) used the top third cutoff [4,33]. A random cutoff was used in nearly half of the studies (10; 43.5%) [6,22,23,25,26,27,28,30,34,36]. With respect to gynecologic surgery, the two studies on endometrial cancer used >12.8 and >50 cases a year as cutoffs [4,6]. Two studies used the top decile cutoff for minimally invasive radical hysterectomy for cervical cancer (>4 cases a year) and minimally invasive oophorectomy for ovarian cancer (>2 cases a year) [5,7].

### 3.5. Perioperative Outcomes

Nearly half of the studies reported a volume–outcome association for perioperative outcomes such as perioperative morbidity, hospital mortality, length of hospital stay, surgical cost, rate of laparotomy conversion, and rate of positive surgical margin (five surgical types in 11 studies) (Table 2) [5,8,29,30,31,32,33,34,35,36,37].

The majority of surgeries were urologic (five studies on minimally invasive radical prostatectomy and two studies on minimally invasive nephrectomy) [30,31,32,33,34,35,36]. In two studies, there was also a volume–perioperative outcome association in pancreaticoduodenectomy [8,29].

In 10 other studies involving three surgical procedures, there was an inconsistent volume–perioperative outcome association (Table 2) [4,6,7,20,22,24,25,26,27,28]. For example, five of the seven studies examining minimally invasive colectomy showed improved perioperative surgical morbidity at high-volume centers [20,22,24,25,28], whereas two studies showed no association [26,27]. Two procedures showed no association between hospital surgical volume and perioperative outcomes (minimally invasive gastrectomy and esophagectomy) [21,23].

Specific to gynecologic surgeries, the two studies on conventional minimally invasive radical hysterectomy for cervical cancer and minimally invasive oophorectomy for ovarian cancer reported an improved surgical morbidity in high-volume centers (Table 2) [5,7]. In contrast to conventional minimally invasive radical hysterectomy, robotic-assisted minimally invasive radical hysterectomy showed no association between hospital surgical volume and perioperative surgical morbidity [7]. Among the two studies on endometrial cancer, one study showed a decreased treatment cost in the high-volume group [6] and the other study showed no association with perioperative outcome [4].

In the sensitivity analysis (Appendix A), the association between hospital surgical volume and the rate of infection or the length of stay was determined. All studies showed that a high hospital surgical volume is associated with shorter lengths of stay, while none found an effect on the rate of infection. We only retrieved one limited study on the effect of hospital surgical volume on the rate of re-operation and cost of hospital stay; notably, the age-specific effects of hospital surgical volume were not analyzed.

### 3.6. Oncologic Outcomes

Only two (8.7%) studies examined the survival outcome related to surgical volume for minimally invasive oncologic surgeries (Table 2) [8,21]. In one study on hepatobiliary surgery reported in 2018, a higher hospital surgical volume for pancreaticoduodenectomy performed for pancreatic cancer was associated with a higher three-year overall survival rate (hazard ratio per one case: 0.98; 95% confidence interval: 0.97–0.99) [8]. Another 2019 study showed that in ≥20 cases (top quartile) of esophagectomies per year for esophageal cancer, there was no association with improved overall survival [21]. Concerning gynecologic malignancies, there were no studies examining the association between MIS volume and survival outcome.

### 3.7. Robotic-Assisted Minimally Invasive Surgery

Among the 23 studies, more than half of the reported volume–outcome associations were related to robotic-assisted MIS (13 studies; 56.5%). These were reported in more recent years from 2012 to 2020 (Appendix A) [6,7,8,20,21,25,29,30,31,32,33,34,35,37]. Nine studies did not mention the number of robotic-assisted MISs [4,5,22,23,24,26,27,28,36].

The association between the study period and volume–outcome association was examined in 16 studies, including 12 studies on robotic-assisted MIS conducted in the United States (Figure 3) [4,5,6,7,8,20,21,24,25,29,30,32,33,34,35,37].

Specifically, the volume–outcome association was assessed because it is temporarily related to the US Food and Drug Administration (FDA) approval of robotic-assisted MIS in each specialty. Robotic-assisted MIS was approved by the FDA in July 2000 for general surgery, March 2001 for thoracic surgery, June 2001 for urologic surgery, and April 2005 for gynecologic surgery [7,38,39]. On average, the time interval from FDA approval to the starting period for the gastrointestinal and urologic studies appears to be longer than that for gynecologic studies.

## 4. Discussion

The key findings of this study are that (i) the association between hospital surgical volume and oncologic outcomes was only investigated in two studies, neither of which included the field of gynecology, and (ii) the effect of hospital surgical volume on perioperative outcomes may be more significant in studies with longer time intervals between the study period and FDA approval.

The fundamental concept of the volume–outcome association was originally proposed in 1979 to regionalize patient care after complex surgical procedures and to improve surgical outcomes [7]. Recent studies have suggested that patient factors and hospital characteristics influence the perioperative outcome [4]. Although the volume–outcome association for perioperative and survival outcomes may be difficult to assess, it has been observed in several surgical procedures [6]. Several volume–outcome associations for perioperative and survival outcomes are well-established and indicate that hospital surgical volume has a major effect on perioperative outcomes in highly complex open surgeries such as pancreaticoduodenectomy, transplantations (heart, lung, liver, and pancreas), and brain tumor resections [40].

However, the reported volume–outcome associations are inconsistent in less complex surgeries [4,41], suggesting that the complexity of the surgical procedure may play a major role in the presence of these associations. This inconsistency may also be a reflection of the lack of consensus on what qualifies a hospital as being high volume. These differences may cause a bias in the volume–outcome associations reported. Therefore, a uniform definition of a high-volume center may aid in accurately determining the volume–outcome associations for perioperative and survival outcomes.

Compared to open surgery, MIS—especially robotic-assisted MIS for malignant diseases—is relatively new [7]. Therefore, little is known about the effects of hospital surgical volume on perioperative and oncologic outcomes in these cases. Our systematic review revealed that the association between hospital surgical volume and perioperative outcomes was inconsistent among studies [4,5,6,7,8,20,21,22,23,24,25,26,27,28,29,30,31,32,33,34,35,36,37]. For gynecologic MIS, the association between hospital surgical volume and perioperative outcomes appears to be modest [4,5,6,7]. Factors contributing to this inconsistency may include (i) the difficulty of the surgical procedure, (ii) differences in the definition of high-volume centers, and (iii) insufficient surgeon experience with the procedure such that a volume–outcome association may not be demonstrable.

Our prior study investigated the association between hospital surgical volume and perioperative outcomes in different surgical approaches (open, conventional MIS, and robotic-assisted MIS for cervical cancer) [7]. In that study, volume–outcome associations were observed for open radical hysterectomy and conventional minimally invasive radical hysterectomy but not for robotic-assisted minimally invasive radical hysterectomy [7]. Open radical hysterectomy has been the standard approach for the surgical treatment of early-stage cervical cancer for several decades. Although it is a rare procedure, conventional minimally invasive radical hysterectomy has also been performed since the early 1990s [7].

In contrast, robotic-assisted minimally invasive radical hysterectomy is a relatively new surgical procedure, and our study reviewed the early experience for robotic-assisted MIS in the United States (Figure 3) [7]. We suggest that the absence of volume–outcome associations in this group could be due to the early learning curve and inadequate experience of using the procedure. This hypothesis is supported by the volume–outcome associations observed for other types of robotic-assisted cancer surgeries. Most studies investigated the effect of hospital surgical volume in robotic-assisted MIS 6–15 years after FDA approval, whereas studies examining robotic-assisted minimally invasive radical hysterectomy were conducted within 3–5 years after FDA approval (Figure 3).

Several studies have identified an association between hospital surgical volume and oncologic outcomes in malignant diseases, including gynecologic cancers [42]. In contrast to open surgery, only two studies investigated the association between hospital surgical volume and the oncologic outcome for MIS [8,21]. Although several studies have shown a feasible oncologic outcome in MIS for malignant diseases, a recent randomized control study showed that minimally invasive radical hysterectomy was associated with lower rates of disease-free survival and overall survival than open radical hysterectomy among women with early-stage cervical cancer (LACC trial) [43]. The effects of hospital surgical volume and surgical skill experience on oncologic outcomes were not investigated in the LACC trial [43]. The aforementioned FDA approval for the use of a robotic-assisted surgical platform was for perioperative outcomes, and the organization encourages researchers to gather more data on the treatment of cancer [44].

To the best of our knowledge, this is the first systematic review to investigate the association between hospital surgical volume and perioperative and oncologic outcomes in MIS. Considering the growing evidence supporting the influence of hospital surgical volume on perioperative outcomes, our study is valuable because it describes the current evidence for MIS.

This study has several limitations. First, this review was limited by the quality and quantity of published evidence. Since MIS is a relatively new procedure, the studies investigating the volume–outcome association for perioperative and survival outcomes are limited. Moreover, the observational studies had classification bias due to differences in the definition of high-volume centers. Second, few studies have investigated the volume–outcome association for perioperative outcomes in gynecologic cancer, thus causing our analysis to be underpowered.

Third, the number of conventional MIS cases could not be identified in approximately half of the studies (unclear number of robotic-assisted MISs); thus, we could not investigate the specific volume–outcome association for the perioperative and survival outcomes of a conventional MIS approach. Fourth, this review may not have included all unpublished studies on the volume–outcome association for perioperative and survival outcomes in MIS, and the underreporting of negative results may have also introduced bias. Due to the limited number of studies, we were unable to find multiple studies with the same conditions (e.g., cancer stage, cancer type, use of conventional MIS, use of robotic-assisted MIS, and area of study).

Fourth, since approximately 70% of the studies were from the United States, it is unclear whether similar results have been observed elsewhere. Finally, although we attempted to examine the effect of hospital surgical volume on the rate of re-operation and cost of hospital stay, only one limited study was retrieved. Moreover, age-specific effects of hospital surgical volume were not analyzed. Further studies are warranted to investigate the effects of hospital volume.

## 5. Conclusions

Our study highlights that the volume–outcome association for oncologic outcome in MIS for gynecologic cancers is not well-established. The available gynecologic studies were performed in the mid to late 2000s. The inconsistency of the volume–outcome associations may be due to the early learning curve and inadequate experience. This lack of evidence calls for further studies to assess the volume–outcome association related to MIS for gynecologic malignancies.

## Figures and Tables

**Figure 1 jcm-10-04787-f001:**
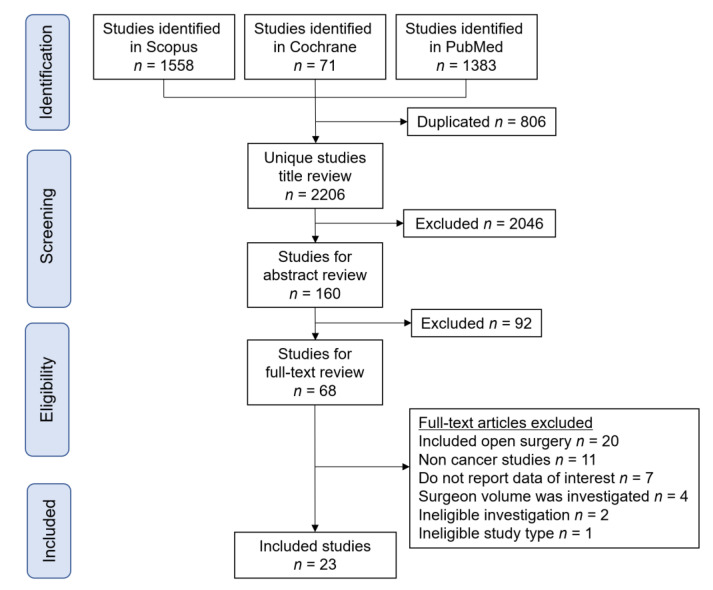
Selection schema for the systematic review of the literature.

**Figure 2 jcm-10-04787-f002:**
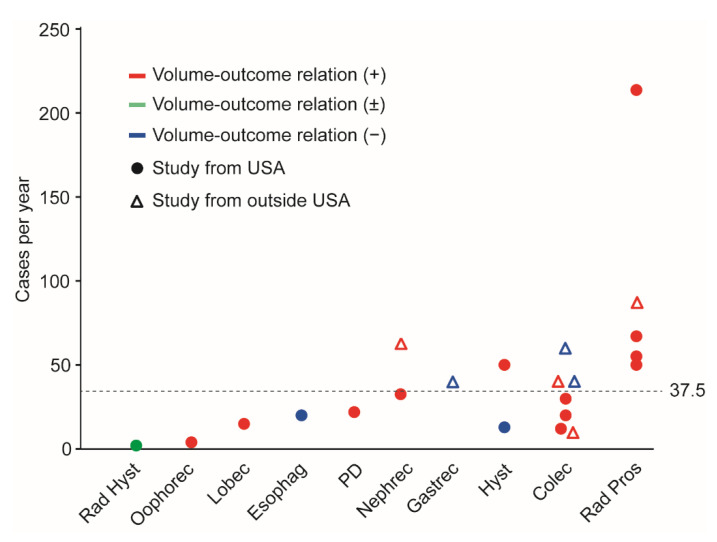
Association between annual hospital surgical cases and volume–outcome. The association between the hospital annual cases of minimally invasive surgery and perioperative outcomes is shown. The median number of annual cases was 37.5. Red indicates the observed volume–outcome association. Green indicates that the volume–outcome association was inconsistent. Blue indicates that the volume–outcome association was not observed. Abbreviations: USA, United States of America; Rad Hyst, radical hysterectomy; oophorec, oophorectomy; lobec, lobectomy; esophag, esophagectomy; PD, pancreaticoduodenectomy; nephrec, nephrectomy; gastrec, gastrectomy; hyst, hysterectomy; colec, colectomy; and Rad Pros, radical prostatectomy.

**Figure 3 jcm-10-04787-f003:**
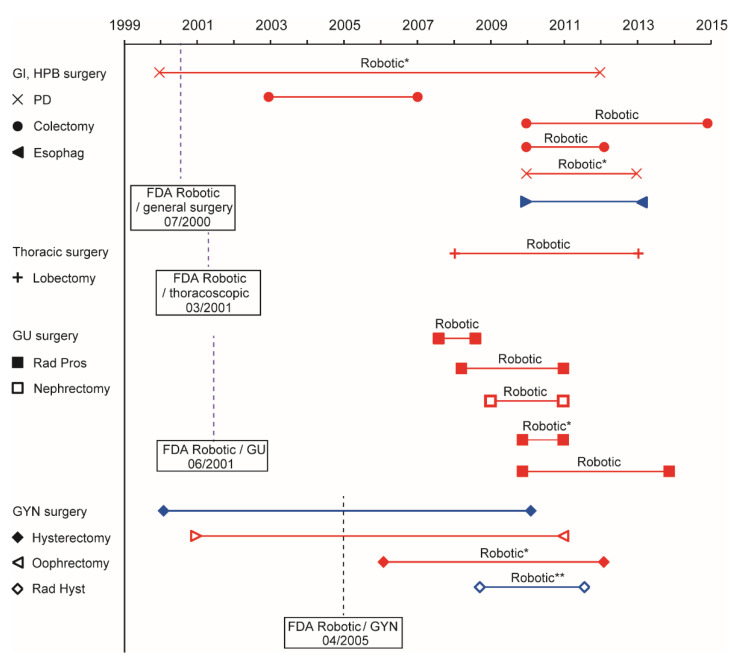
Association between the study period and volume–outcome. Studies from United States are included in this figure. * Mixed with conventional laparoscopic surgery. ** Cases with conventional laparoscopic surgery were excluded from this analysis. Red shapes and lines: the volume–outcome association for perioperative outcomes was observed. Blue shapes and lines: the volume–outcome association was not observed. Abbreviations: FDA Robotic, The Food and Drug Administration approval for robotic-assisted minimally invasive surgery; Robotic, robotic surgery; GI, gastrointestinal; HPB, hepato–pancreato–biliary, GU, genitourinary; GYN, gynecology; PD, pancreaticoduodenectomy; Esophag, esophagectomy; Rad Pros, radical prostatectomy; and Rad Hyst, radical hysterectomy.

**Table 1 jcm-10-04787-t001:** Definition of high-volume center.

Author	Year	Study Period	Category	Cancer Type	Surgery Type	HV def (/yr)	HV Classification
Matsuo K [5]	2020	2001–2011	GYN	Ovarian Ca	Oophorectomy	>2	90%ile
Matsuo K [7]	2020	2007–2011	GYN	Cervical Ca	Radical hysterectomy	>4	90%ile
Wright JD [6]	2014	2006–2012	GYN	EM Ca	Hysterectomy	>50	Random
Wright JD [4]	2012	2000–2010	GYN	EM Ca	Hysterectomy	>12.8	Top 3rd
Concors SJ [20]	2019	2010–2015	GI	Colorectal Ca	Colectomy	≥12	90%ile
Gietelink L [22]	2016	2011–2012	GI	Colorectal Ca	Colectomy	≥40	Random
Zheng Z [24]	2014	2003–2007	GI	Colorectal Ca	Colectomy	≥30	QT1
Keller DS [25]	2013	2010–2012	GI	NA	Colectomy	>20	Random
Kuwabara K [26]	2009	2007	GI	Colorectal Ca	Colectomy	≥60	Random
Yasunaga H [27]	2009	2006–2007	GI	Colorectal Ca	Colectomy	≥40	Random
Kuhry E [28]	2005	1997–2003	GI	Colorectal Ca	Colectomy	≥10	Random
Murata A [23]	2015	2009–2011	GI	Gastric Ca	Gastrectomy	≥40	Random
Salfity H [21]	2019	2010–2013	GI	Esophag Ca	Esophagectomy	≥20	QT1
Nassour I [8]	2018	2010–2013	HPB	Pancreas Ca	PD	NA	^¶^
Adam MA [29]	2017	2000–2012	HPB	Pancreas Ca	PD	>22	RCSs *
Xia L [30]	2020	2010–2014	GU	Prostate Ca	Radical prostatectomy	≥219	Random
Weiner AB [32]	2015	2010–2011	GU	Prostate Ca	Radical prostatectomy	>72	QT1
Hyams ES [34]	2013	2008–2011	GU	NA	Radical prostatectomy	>60	Random
Yu HY [35]	2012	2008	GU	Prostate Ca	Radical prostatectomy	≥55	QT1
Budäus L [36]	2011	2005–2008	GU	Prostate Ca	Radical prostatectomy	≥92	Random
Peyronnet B [31]	2018	2009–2015	GU	Renal Ca	Partial nephrectomy	>70	QT1
Monn MF [33]	2014	2009–2011	GU	Renal tumor	Partial nephrectomy	≥35	Top 3rd
Tchouta LN [37]	2017	2008–2013	Other	Lung (NA)	Lobectomy	≥15	QT1

^¶^ Surgical volume was analyzed with continuous variables. * Restricted cubic splines (RCSs) were used to specify and estimate the functional form of the annual hospital surgical volume with respect to the incidence of any complication. Abbreviations: NA, not applicable; HV, high-volume center; def, definition; GYN, gynecology; GI, gastrointestinal, HPB, hepato–pancreato–biliary; GU, genitourinary; Ca, cancer; PD, pancreaticoduodenectomy; EM, endometrial; Esophag, esophageal; QT1, top quartile; random, random number of cases; and yr, year.

**Table 2 jcm-10-04787-t002:** Volume–outcome relationship in minimally invasive surgeries for malignant diseases.

Surgery Type	Category	Robotic	Author	Year	No.	HV (/yr)	Surgical Outcome	Oncologic Outcome
Volume–outcome relationship, observed								
Oophorectomy	GYN	-	Matsuo [5]	2020	4822	>4	↓complication	--
PD	HPB	Yes *	Nassour [8]	2018	1623	^¶^	--	↑3-year OS
	HPB	Yes *	Adam [29]	2017	865	>22	↓complication	--
Radical prostatectomy	GU	Yes	Xia [30]	2020	114,957	≥219	↓length of stay, ↓PSM	--
	GU	Yes *	Weiner [32]	2015	87,415	>72	↓lap conversion	--
	GU	Yes	Hyams [34]	2013	1489	>60	↓surgical cost	--
	GU	Yes	Yu [35]	2012	2348	≥55	↓complication	--
	GU	-	Budäus [36]	2011	2108	≥92	↓length of stay	--
Nephrectomy **	GU	Yes	Peyronnet [31]	2018	1222	>70	↓complication, ↓PSM	--
	GU	Yes	Monn [33]	2014	17,583	≥35	↓complication	--
Lobectomy	Other	Yes	Tchouta [37]	2017	8253	≥15	↓hospital mortality	--
Volume–outcome relationship, inconsistent								
RH	GYN	Yes *	Matsuo [7]	2020	2202	>2	LSC: ↓complication	--
							Robotic: no association	--
Hysterectomy	GYN	Yes *	Wright [6]	2014	10,906	>50	↓cost ^‡^	--
	GYN	-	Wright [4]	2012	4137	>12.8	No association	--
Colectomy	GI	Yes	Concors [20]	2019	8107	≥12	↓lap conversion, ↓PSM	--
	GI	-	Gietelink [22]	2016	5161	≥40	↓PSM	--
	GI	-	Zheng [24]	2014	4617	≥30	↓length of stay, ↓hospital mortality	--
	GI	Yes	Keller [25]	2013	1428	>20	↓complication	--
	GI	-	Kuwabara [26]	2009	3765	≥60	No association	--
	GI	-	Yasunaga [27]	2009	1212	≥40	Complication: no association	--
	GI	-	Kuhry [28]	2005	627	≥10	↓resp complication	--
Volume–outcome relationship, not observed								
Gastrectomy	GI	-	Murata [23]	2015	5941	≥40	Complication: no association	--
Esophagectomy	GI	No	Salfity [21]	2019	2371	≥20	Mortality: no association	No associationfor OS

^¶^ Surgical volume was analyzed using continuous variables. * Mixed with conventional laparoscopic surgery. ** Partial nephrectomy. ^‡^ Cost data represent the cost of the entire index hospitalization. Abbreviations: -, not specified; --, not assessed; No., number; yr, year; NS, not specified; HV, definition of high-volume center; GYN, gynecology; HPB, hepato–pancreato–biliary; GI, gastrointestinal, GU, genitourinary; PD, pancreaticoduodenectomy; RH, radical hysterectomy; GI, gastrointestinal; PSM, positive surgical margin; lap, laparotomy; resp, respiratory; LSC, laparoscopic surgery; robotic, robotic surgery; and OS, overall survival.

## Data Availability

All the studies used in this study are published in the literature.

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
