# Peer review of "Minimally Invasive Surgery and Surgical Volume-Specific Survival and Perioperative Outcome: Unmet Need for Evidence in Gynecologic Malignancy"

_jcm, 2021, doi:10.3390/jcm10204787_

Round 1

Reviewer 1 Report

Dear authors, I find the manuscript in this final format very interesting and of great methodological quality.

While it is true that all gynecological oncologists have opted for minimal invasion, it is also true that we have made little effort to check the quality of our surgeries. Instead we have endeavored to concentrate as much cases as possible in order to increase our surgical skills. Nothing is further from the truth, you have shown with your systematic review that the benefit of this concentration of number of surgeries in our hospitals is not entirely clear, that we must now investigate what is the reality once the learning curve of the laparoscopy in oncology is done.

I can only congratulate you and recommend the publication of your manuscript in this great journal.

Reviewer 2 Report

Thank you for submitting the edited version of the manuscript after incorporation of the suggestions.

This manuscript is a resubmission of an earlier submission. The following is a list of the peer review reports and author responses from that submission.

Round 1

Reviewer 1 Report

The submitted study examined the effect of hospital surgical volume on oncologic outcomes in minimally invasive surgery (MIS) for gynecologic malignancies. The authors performed an intensive literature search and represent the data in a well informative way. However, I have certain comments regarding this study which need to be included in the manuscript.

  1. What is the possible percentage of hospital acquired infection due to MIS? Need to include related data in the manuscript.
  2. What is the possible redo number of cases performed using MIS and what is the length of gap between first MIS to second MIS?
  3. What is the average length of stay of patient using MIS and its correlation with cost of stay in the hospital?
  4. Were any specific age group patients are studied who underwent MIS need to include age wise data in the manuscript?

Therefore, by incorporating data related to the comments will improve the manuscript readability.

Author Response

Reviewer #1

The submitted study examined the effect of hospital surgical volume on oncologic outcomes in minimally invasive surgery (MIS) for gynecologic malignancies. The authors performed an intensive literature search and represent the data in a well informative way. However, I have certain comments regarding this study which need to be included in the manuscript.

Reply: Thank you for your positive comments. To make the manuscript more informative, we have revised it according to the reviewers’ comments.

Reviewer #1, comment 1

What is the possible percentage of hospital acquired infection due to MIS? Need to include related data in the manuscript.

Reply: Lines 119-122, line 225, lines 351-355, Supplemental Table S4

Thank you for your suggestion. We reviewed the included studies again and found that 7 of the 23 studies examined the effect of hospital surgical volume (HSV) on the rate of infection. All studies showed that HSV is not associated with a decreased rate of infection.

Reviewer #1, comment 2

What is the possible redo number of cases performed using MIS and what is the length of gap between first MIS to second MIS?

Reply: Lines 119-122, line 226, lines 351-355, Supplemental Table S4

In this systematic review, only one study investigated the effect of HSV on the rate of re-operations. We have addressed this potential limitation of this study in the Discussion.

Reviewer #1, comment 3

What is the average length of stay of patient using MIS and its correlation with cost of stay in the hospital?

Reply: Lines 119-122, line 224, lines 351-355, Supplemental Table S4

We appreciate the reviewer’s valuable comments. We reviewed the included studies again, finding that 10 of 23 studies examined the influence of HSV on the length of hospital stay, with all demonstrating a positive relationship between them.

Reviewer #1, comment 4

Were any specific age group patients are studied who underwent MIS need to include age wise data in the manuscript?

Reply: Lines 119-122, line 227, lines 351-355, Supplemental Table S4

None of the studies included in this systematic review study investigated the effect of HSV according to age. We have addressed this potential limitation of this study in the revised Discussion.

Therefore, by incorporating data related to the comments will improve the manuscript readability.

Reviewer 2 Report

In the first instance I would like to congratulate the authors for this systematic review that I find very interesting.
secondly, I find a series of methodological considerations lacking that a review of this quality deserves to have:
In the methodology section, when the inclusion and exclusion criteria are defined, the question PICCO (Population, Intervention, Comparison, Outcomes), recommended by the PRISMA guidelines, must be well defined.1

1.- van Loveren C, Aartman IH. De PICO-vraag [The PICO (Patient-Intervention-Comparison-Outcome) question]. Ned Tijdschr Tandheelkd. 2007 Apr;114(4):172-8. Dutch. PMID: 17484414.

I would also like the authors to define the tool they have used and its detailed description to assess the quality of the studies included in the review and to make a table with the result of the assessment of the bias as supplementary material.

Regarding the discussion and conclusions, I totally agree that the relationship between the hospital volume and the results of the surgery in terms of the minimally invasive approach has not been sufficiently studied. Perhaps the minimally invasive techniques are too recent, but the What is clear is that we have adopted this approach in oncology without having designed randomized prospective studies comparing classical techniques with these more recent ones and much less comparing the results in terms of survival.
Systematic reviews always generate hypotheses for new research, perhaps this article is the starting point to design prospective studies that assess not only the volume of cases per hospital / surgeon but also the long-term result for the patient.
I can only congratulate the authors for this work.

Author Response

Reviewer #2

In the first instance I would like to congratulate the authors for this systematic review that I find very interesting.

Thank you for your positive comments on the manuscript. We have revised the main text according to the reviewer’s comments.

Reviewer #2, comment 1

secondly, I find a series of methodological considerations lacking that a review of this quality deserves to have:

In the methodology section, when the inclusion and exclusion criteria are defined, the question PICCO (Population, Intervention, Comparison, Outcomes), recommended by the PRISMA guidelines, must be well defined.1

1.- van Loveren C, Aartman IH. De PICO-vraag [The PICO (Patient-Intervention-Comparison-Outcome) question]. Ned Tijdschr Tandheelkd. 2007 Apr;114(4):172-8. Dutch. PMID: 17484414.

Reply: Lines90-91, Reference No.14, Supplemental Table S1

We appreciate the reviewer’s insightful comments. According to the reviewer’s suggestion, we have prepared a table to demonstrate how the PICO framework was utilized and cited the suggested study.

Reviewer #2, comment 2

I would also like the authors to define the tool they have used and its detailed description to assess the quality of the studies included in the review and to make a table with the result of the assessment of the bias as supplementary material.

Reply: Lines 117-118, lines 136-139, Supplemental Table S3

Thank you for your helpful comments. We have described how we assessed the quality of the included studies.

Regarding the discussion and conclusions, I totally agree that the relationship between the hospital volume and the results of the surgery in terms of the minimally invasive approach has not been sufficiently studied. Perhaps the minimally invasive techniques are too recent, but the What is clear is that we have adopted this approach in oncology without having designed randomized prospective studies comparing classical techniques with these more recent ones and much less comparing the results in terms of survival.

Systematic reviews always generate hypotheses for new research, perhaps this article is the starting point to design prospective studies that assess not only the volume of cases per hospital / surgeon but also the long-term result for the patient.

I can only congratulate the authors for this work.

We appreciate your positive comments.
